# The Novel A-Type Proanthocyanidin-Rich Phytocomplex SP4™ Acts as a Broad-Spectrum Antiviral Agent against Human Respiratory Viruses

**DOI:** 10.3390/ijms25137370

**Published:** 2024-07-05

**Authors:** Giulia Sibille, Giuseppe Mannino, Ilaria Frasson, Marta Pavan, Anna Luganini, Cristiano Salata, Massimo E. Maffei, Giorgio Gribaudo

**Affiliations:** 1Microbiology and Virology Unit, Department of Life Sciences and Systems Biology, University of Torino, Via Accademia Albertina 13, 10123 Turin, Italy; giulia.sibille@unito.it (G.S.); marta.pavan@unito.it (M.P.); anna.luganini@unito.it (A.L.); 2Plant Physiology Unit, Department of Life Sciences and Systems Biology, University of Torino, Via Quarello 15/a, 10135 Torino, Italy; giuseppe.mannino@unito.it (G.M.); massimo.maffei@unito.it (M.E.M.); 3Department of Molecular Medicine, University of Padova, 35121 Padova, Italy; ilaria.frasson@unipd.it (I.F.); cristiano.salata@unipd.it (C.S.)

**Keywords:** natural antivirals, broad-spectrum antiviral activity, coronaviruses, influenza virus, respiratory syncytial virus, SP4™ phytocomplex, A-type proanthocyanidins, virucidal activity

## Abstract

The appearance of new respiratory virus infections in humans with epidemic or pandemic potential has underscored the urgent need for effective broad-spectrum antivirals (BSAs). Bioactive compounds derived from plants may provide a natural source of new BSA candidates. Here, we investigated the novel phytocomplex formulation SP4™ as a candidate direct-acting BSA against major current human respiratory viruses, including coronaviruses and influenza viruses. SP4™ inhibited the in vitro replication of SARS-CoV-2, hCoV-OC43, hCoV-229E, Influenza A and B viruses, and respiratory syncytial virus in the low-microgram range. Using hCoV-OC43 as a representative respiratory virus, most of the antiviral activity of SP4™ was observed to stem primarily from its dimeric A-type proanthocyanidin (PAC-A) component. Further investigations of the mechanistic mode of action showed SP4™ and its PAC-A-rich fraction to prevent hCoV-OC43 from attaching to target cells and exert virucidal activity. This occurred through their interaction with the spike protein of hCoV-OC43 and SARS-CoV-2, thereby interfering with spike functions and leading to the loss of virion infectivity. Overall, these findings support the further development of SP4™ as a candidate BSA of a natural origin for the prevention of human respiratory virus infections.

## 1. Introduction

One of the many hard lessons that the COVID-19 pandemic taught the world concerns the importance of having an arsenal of effective antiviral drug interventions ready to deploy in the face of emerging respiratory virus infections to prevent their progression to acute disease forms [1,2,3]. Indeed, the shortage of broad-spectrum antiviral (BSA) drugs severely impacted the world’s response to the pandemic in its first year. Had they been available, the pandemic of SARS-CoV-2 might have been significantly different, and the number of hospitalizations and deaths dramatically reduced. Initiating treatment with BSAs as soon as possible after virus emergence may also provide more time for the development of new virus-specific vaccines, monoclonal antibodies, and antiviral drugs, as well as for implementing non-pharmaceutical countermeasures to slow the spread of infections [4,5]. Accordingly, new BSAs are urgently needed to combat future respiratory tract virus infections (RTIs), with novel coronaviruses and influenza viruses being the most likely to have pandemic potential [6]. Candidate BSAs should be prospectively identified and developed up to human safety studies, ready for clinical studies in infected individuals upon the outbreak of a new virus, rendering their authorization and subsequent deployment as fast as possible [4,5,7].

BSA compounds should be able to interfere with the replication of multiple viruses, belonging to different virus families, due to their ability to target similar structures and/or the biochemical pathways which the viruses employ to synthetize their components and replicate in host cells [8]. However, given the substantial diversity in viral structures and replication strategies, the development of effective BSAs has proven to be more challenging than that of most direct-acting antivirals (DAAs), which target the activity of a virus-specific polymerase or protease. To identify new BSA targets, it is first necessary to identify the constraining protein functions or the critical cellular processes common to different viruses.

The increasing number of new viral outbreaks in the last two decades has reiterated the burning need for molecules able to implement the “one drug, multiple viruses” paradigm. New oral BSAs, in addition to a high potency, must also be endowed with low cytotoxicity and a high tolerability to facilitate patient compliance, drug administration, and their access by even remote and low-income countries. Such drugs would be of undeniable value in the fight against future respiratory viral threats with pandemic potential [2,4,5].

Given the great challenge that the development of effective BSAs presents, natural products have also been considered, being a great source of chemical complexity and diversity [9]. In fact, an increasing body of evidence demonstrates the antiviral activity of plant-derived extracts against some common human respiratory viruses, advocating their potential for further development into novel BSAs [10,11]. Polyphenols, flavonoids, glycosides, terpenes, and alkaloids have all been identified as the bioactive components of plant extract-derived antivirals, and the mechanisms of action and molecular targets for some of these phytoconstituents have already been elucidated [12,13]. Of particular interest are the A-type proanthocyanidins (A-type PACs, PAC-As), a large group of natural polyphenols, occurring as oligomers or polymers of flavan-3-ol linked by an unusual double linkage. PAC-As have been identified as major contributors to the antibacterial and antiviral activities of some of the plant extracts from which they were originally isolated and chemically characterized [14,15].

On the basis of these premises, the aim of this study was to evaluate the BSA potential of the innovative PAC-A-rich phytocomplex SP4™ against human respiratory viruses. SP4™ is a new formulation designed to contain four different plant extracts obtained from recycled materials from the food industry, and each extract is characterized by its high content of PAC-As and other polyphenols [16]. SP4™ has been observed to exert antioxidant activity and prevent bacterial adhesion of P-fimbriated uropathogenic *Escherichia coli* [16]. Here, we report on the ability of SP4™ to inhibit the in vitro replication of SARS-CoV-2 and other human endemic coronaviruses, as well as that of influenza viruses A and B and respiratory syncytial virus. Dimeric PAC-As (see Appendix A) were identified as the phytochemical constituents of SP4™ responsible for most of its antiviral activity. Finally, we show its main mechanism of action against a representative hCoV to involve the alteration in the spike envelope glycoprotein function, in turn inhibiting viral attachment. Together, these results show SP4™ to be a highly attractive candidate molecule from which to develop novel BSAs for the prevention of human respiratory infections.

## 2. Results

### 2.1. SP4™ Exerts a Broad-Spectrum Antiviral Activity against Human Respiratory Viruses

To investigate the possible antiviral activity of SP4™ against representative human respiratory viruses, we performed two different antiviral assays (namely, focus forming reduction assays (FFRAs) performed in HCT-8 and MRC5 cells, and virus yield reduction assays (VRAs) performed in Calu-3 and A549 cells) using the coronaviruses hCoV-OC43, hCoV-229E, and SARS-CoV-2, influenza A and B viruses (IAV and IBV), and the respiratory syncytial virus (RSV). As shown in Figure 1A–D, full-treatment with SP4™ (i.e., applied before, during, and after infection) exhibited a remarkable concentration-dependent inhibitory effect on coronavirus replication.

The calculated 50% effective concentration (EC_50_) and 90% effective concentration (EC_90_) value were in the low-microgram range (see Table 1). Importantly, the low 50% cytotoxic concentration (CC_50_) of SP4™ as determined in uninfected cells confirmed that the antiviral activity of SP4™ did not stem from non-specific cytotoxicity. Indeed, the selectivity index (SI) was greater than 60, 110, and 20 for hCoV-OC43, hCoV-299E, and SARS-CoV-2, respectively (Table 1). 

As depicted in Figure 1E–G, the VRAs performed in A549 cells showed SP4™ to suppress the replication of IAV, IBV, and RSV-A, confirming its antiviral activity also against these human respiratory viruses. The EC_50_ and EC_90_ values were, again, in the low-microgram range (Table 1), and the positive SI (Table 1) indicated that the anti-IV and anti-RSV activity of SP4™ was not due to a negative effect on cell viability. 

These results confirm SP4™ to exert a broad-spectrum antiviral activity against a range of important human respiratory viruses.

### 2.2. PAC-As Are Responsible for the Antiviral Activity of SP4™

To pinpoint the bioactive compound(s) in SP4™ responsible for its antiviral activity, we subjected the phytocomplex to flash chromatography (FC) to obtain its three primary fractions, namely F1, F2, and F3. We then applied high-performance liquid chromatography–diode array detection–electro-spray ionization tandem mass spectrometry (HPLC-DAD-ESI-MS/MS) to identify the chemical components of these fractions. F1 was mostly composed of flavonoids, including quercetin and its methyl and dihydro glycoside derivatives, catechin, and epicatechin. This fraction also contained a very small amount of PAC-A3 and PAC-Bs (Appendix A).

Further separation of F1 by FC yielded three secondary fractions (F1A, F1B, and F1C). F1A was dominated by dihydroquercetin glycosides; F1B had a high content of quercetin and myricetin glycosides, catechin and epicatechin, and a small amount of PAC-B1; and F1C contained primarily flavonoid aglycones, including quercetin, myricetin, and their glucoside and dihydro derivatives (Appendix A). F2 contained mainly anthocyanins and anthocyanidin, and it was not fractionated any further. F3 was characterized by the presence of very high amounts of PAC-As, such as PAC-A1, PAC-A2, and PAC-A4, and other PAC-Bs with different degrees of polymerization (DP) (Appendix A). The sub-fractionation of F3 by FC yielded five secondary fractions (F3A, F3B, F3C, F3D, and F3E). F3A contained primarily PAC-As and PAC-Bs of different DP. F3B contained mostly polymeric PAC-Bs, while the composition of F3C was similar to that of F3A but with a higher content of dimeric PAC-A1, PAC-A2, and PAC-A4. F3D produced a composition similar to that of F3A, but with a lower amount of PAC-As. Finally, F3E contained only very low amounts of polymeric PAC-Bs (Appendix A).

The purified primary and secondary fractions of SP4™ were then tested in antiviral assays against hCoV-OC43, selected as a representative human respiratory virus, the replication of which was strongly impaired by the phytocomplex (Figure 1A). Analysis of the anti-hCoV activity of the primary fractions F1, F2, and F3 revealed F3 to contain the components responsible for the main inhibitory activity of SP4™ against hCoV-OC43 (Figure 2A). As shown in Table 2, the EC_50_ values indicated F3 to possess the highest inhibitory activity, followed by F1. F2 exhibited only a minimal inhibitory activity.

The fractionation of F1 allowed us to identify F1B as the prime fraction with anti-hCoV-OC43 activity (Figure 2B). Indeed, as shown in Table 2, the EC_50_ value for F1B was the lowest among the secondary F1-derived fractions. By contrast, as depicted in Figure 2C, all the secondary fractions obtained from F3 exerted an inhibitory effect on hCoV-OC43 replication, especially F3A, F3B, and F3C. Noteworthy is the fact that F3C had the lowest EC_50_, at 0.31 ± 0.15 μg/mL (Table 2), and was also the richest in dimeric PAC-As, such as PAC-A1, PAC-A2, PAC-A3, and PAC-A4 (Appendix A). The chemical structures of these dimeric, bioactive PAC-As of SP4™ are shown in Appendix A.

These results suggest the antiviral activity of SP4™ to rely primarily on the phytoconstituents present within F3 and, more specifically, in F3C. Moreover, by considering the results of the antiviral assays together with the qualitative and quantitative chemical analyses reported in Appendix A, we can propose PAC-As to be the bioactive components responsible for most of the antiviral activity of SP4™.

To gather further evidence that the PAC-A components of SP4™ constitute its principal antiviral agents, we tested synthetic dimeric PAC-A1, PAC-A2, and PAC-A4 (Appendix A)—the most abundant PAC-As in F3C (Appendix A)—against hCoV-OC43. As control synthetics, PAC-B1, PAC-B2, and PAC-B3 were also included in the assays, since they were also found in high amounts in F3. As shown in Figure 3A, all synthetic PAC-As inhibited hCoV-OC43 replication in a concentration-dependent manner. Importantly, as reported in Table 3, the EC_50_ values of PAC-A1 and PAC-A2 were higher than those of either SP4™ (Table 1), F3, or F3C (Table 2), thus suggesting a possible synergistic effect occurring between PAC-A1 and PAC-A2.

The antiviral activity of the synthetic PAC-As was greatest in PAC-A1, followed by PAC-A2, and then PAC-A4 (Table 3). Conversely, synthetic PAC-Bs (Figure 3B) exerted no considerable inhibitory activity (the EC_50_ values were all >130 μg/mL) (Table 3).

Taken together, the results of this section point towards dimeric PAC-As as the active component of SP4™ responsible for the greater part of the phytocomplex’s anti-CoV activity.

### 2.3. SP4™ Targets Early Phases in the hCoV Replication Cycle

Next, to characterize which phase of the hCoV-OC43 replicative cycle is targeted by SP4™, time-of-drug-addition experiments were carried out according to the scheme depicted in Figure 4A. 

In brief, HCT-8 cells were exposed to different concentrations of SP4™ prior to hCoV-OC43 infection (from −1 to 0 h, pre-treatment; Pre-T), during viral infection (from 0 to 2 h p.i.; adsorption stage, co-treatment, Co-T), or after virus adsorption (from 2 to 72 h p.i.; post-treatment, Post-T) (Figure 4A). The EC_50_ values obtained indicated SP4™ to be most effective when present during either the pre-adsorption (Pre-T) (Figure 4B, left panel) or infection stages (Co-T) (Figure 4B, middle panel), with EC_50_ values of 3.79 ± 0.036 μg/mL in the Pre-T condition and 4.91 ± 0.020 μg/mL for Co-T. Conversely, SP4™ was much less effective when added during the post-entry stage (Post-T) (Figure 4B, right panel), indicated by its higher EC_50_ of 29.05 ± 1.009 μg/mL. These results clearly suggest the phytocomplex to exert its antiviral activity during the early stages of the hCoV-OC43 replication cycle, which comprises cell attachment and entry.

To investigate further the ability of SP4™ to target the early phases of the hCoV replication cycle, we performed specific temperature-shift assays to differentiate between viral attachment and entry. For the attachment assay, HCT-8 cell monolayers were chilled and then infected with hCoV-OC43 in the presence of different concentrations of SP4™ or the PAC-A-rich F3 for 2 h at 4 °C. These conditions permit the attachment of the viral particles but not their entry into cells. Thereafter, compounds and unattached viral particles were removed, and the remaining hCoV-OC43 infectivity was evaluated after incubation at 33 °C for 72 h. As shown, in Figure 5A, both SP4™ and F3 prevented the attachment of hCoV-OC43 in a concentration-dependent manner. The EC_50_ for SP4™ and F3 were 2.18 ± 0.09 μg/mL and 0.89 ± 0.11 μg/mL, respectively. It is noteworthy that these EC_50_ values were very similar to those obtained in the FFRA assays for SP4™ and F3 (Table 1 and Table 3), suggesting that the anti-hCoV-OC43 activity obtained through full-T could be reproduced via a short exposure (2 h) without any significant loss of potency.

The effect of SP4™ and F3 on the entry phase of hCoV-OC43 was examined by infecting pre-chilled HCT-8 cells at 4 °C for 2 h. The infected cultures were then treated with different concentrations of SP4™ or F3 and shifted to 33 °C for 2 h to permit the entry of attached hCoV-OC43 particles. At the end of this time interval, hCoV viruses still bound to the cells were inactivated by treatment with acidic buffer, and the remaining hCoV-OC43 infectivity was evaluated at 72 h p.i. Figure 5B shows that both SP4™ and F3 were able to inhibit hCoV-OC43 cell entry in a concentration-dependent manner. Here, the measured EC_50_ values were 6.45 ± 0.14 μg/mL for SP4™ and 2.51 ± 0.04 μg/mL for F3. These results tell us that both SP4™ and F3 were more effective at interfering with cell attachment than with the subsequent entry phase of a prototype hCoV replication cycle.

Together, the results of this section sustained the ability of SP4™ or F3, the richest fraction in PAC-As, to preferentially affect virus attachment, therefore suggesting the suitability of SP4™ as an early-acting inhibitor of hCoV infection. 

### 2.4. SP4™ Exerts Virucidal Activity against hCoV-OC43

The interference SP4™ with the attachment phase could be due to its action on hCoV particles prior to their interaction with host cells. Hence, to verify this hypothesis, aliquots of hCoV-OC43 infectious particles were incubated with 15 μg/mL of either SP4™ or F3 for 0, 60, 120, or 180 min at either 4 °C or 33 °C. Thereafter, the mixtures were diluted to reduce the concentration of SP4™ or F3 well below those inhibiting virus attachment (Figure 5), and then the residual hCoV-OC43 infectivity of the diluted samples was then determined by FFRA on HCT-8 cells. 

As shown in Figure 6A, the incubation of hCoV particles with either SP4™ or F3 at 4 °C for longer times did not bring about any further reduction in hCoV infectivity. Conversely, incubation at 33 °C abolished mostly the infectivity of hCoV-OC43 after 120 min of incubation, thus indicating that SP4™ and F3 were able to inactivate infectious hCoV (i.e., demonstrate virucidal activity) in time- and temperature-dependent manners (Figure 6B).

These results, therefore, suggested that the anti-hCoV activity of SP4™ derives from its ability to affect hCoV virions, thus impairing their attachment to target cells.

### 2.5. SP4™ Interacts with the Spike Proteins of hCoV-OC43 and SARS-CoV-2

The spike glycoprotein (S) of coronaviruses is essential for both virus attachment and entry into host cells since it mediates both virus interaction with cell-surface receptors and the fusion between the viral envelope and the cell membranes [17]. 

Since we observed a severe reduction in hCoV-OC43 virions’ infectivity after their incubation with either SP4™ or F3 (Figure 6), we used electrophoretic mobility shift assay experiments [18] to investigate whether SP4™ and the PAC-A-rich F3 fraction were able to affect the spike glycoprotein of hCoV-OC43 and thereby interfere with its functions during cell attachment. To this end, we mixed aliquots of purified recombinant hCoV-OC43 S protein produced in a baculovirus expression system with increasing concentrations of either SP4™ or F3, then incubated at 37 °C. As depicted in Figure 7A (left panel), the fractionation of this recombinant S protein preparation by SDS-PAGE produced a larger albeit less evident band of about 160 kDa, corresponding to the S1 + S2 complex, and two clear bands of about 95 and 61 kDa, corresponding to the S1 and S2 proteins, respectively. However, exposure of the hCoV-OC43 S protein preparation to increasing amounts of either SP4™ (Figure 7B) or F3 (Figure 7C) brought about reductions in the amounts of all the S components, with the greatest reductions observed in the S1 and S2 protein bands at the highest concentrations of SP4™ or F3, together with the appearance of a new protein band of more than 200 kDa which was not present in untreated S protein samples. Similarly, the exposure of S proteins to either SP4™ (Figure 7D) or F3 (Figure 7E) for different times resulted in a reduction in the S components and, again, the appearance of the new heavier protein band, characterized by less electrophoretic mobility compared to the untreated controls.

Subsequently, we incubated a purified recombinant SARS-CoV-2 (variant Omicron BA.1) S glycoprotein with increasing concentrations of either SP4™ or F3. The Omicron BA.1 S protein was expressed as a mutant in the furin cleavage site in HEK293 cells, so as to produce a trimeric structure detectable by SDS-PAGE as a single, monomeric S1 + S2 protein of about 160 kilodaltons (kDa) (Figure 8A). As previously observed for the hCoV-OC43 spike protein, exposing this SARS-CoV-2 S glycoprotein to either SP4™ (Figure 8B) or F3 (Figure 8C) led to a severe concentration-dependent decrease in the intensity of the S1 + S2 protein band and the appearance of other protein bands with less electrophoretic mobility. Moreover, the incubation of recombinant SARS-CoV-2 S with either SP4™ (Figure 8D) or F3 (Figure 8E) for different times once again produced a reduction in the S1 + S2 band and an alteration in its electrophoretic mobility toward forms with higher molecular weights (Figure 7D,E). These findings support the hypothesis that SP4™ and F3 interact with the spike protein of SARS-CoV-2.

Taken together, these results indicate the ability of SP4™ and F3, the fraction richest in A-type PACs, to interact with the spike proteins of two representative human coronaviruses and that these interactions impair spike function during virus attachment to target cells, underlying the overall antiviral activity of SP4™ against hCoVs.

## 3. Discussion

Respiratory tract infections caused by hCoVs, IVs, and RSV are a predominant cause of morbidity and mortality worldwide and present a major health problem for society to address [19]. Despite this, the arsenal of current antiviral drugs against these viruses is limited, conferring major challenges in the therapeutic management of RTIs as a complementary approach to vaccination. These facts clearly sustain the urgent need for new BSAs to deploy against these currently circulating viruses as well as strains emerging from future zoonoses. In the attempt to meet this need, natural products derived from plant extracts can be evaluated as potential sources of BSA candidates able to interfere with the early phases of the replication cycle of different respiratory viruses and offer new possible strategies for the prevention of present and future RTIs. However, this approach does present certain issues that need to be tackled, including the following: the production of highly effective, safe, and standardized plant extracts; the identification of the bioactive components responsible for their antiviral activity; the characterization of their mechanism(s) of action, which is often related to the synergistic cooperation of different constituent components; and the affordability of antiviral plant extracts for low-income countries. The results of our study endorse the eligibility of the SP4™ phytocomplex, rich in A-type PACs, as a direct-acting BSA candidate against respiratory viruses, able to overcome the above-mentioned hurdles.

These findings deserve further consideration, such as the mechanism of the antiviral activity of SP4™. Antiviral activities against several human respiratory viruses have been described for many plant-derived extracts [13,15]. However, PAC-As were identified as bioactive antiviral agents in just a few studies directed at IVs and hCoVs. We have previously reported on the ability of a cranberry extract endowed with a high content of A-type PAC dimers, including PAC-A2 (Appendix A), to inhibit the in vitro replication of both IAV and IBV [18]. Mechanistic studies demonstrated the ability of this cranberry extract to hamper the attachment and entry phases of IVs and exert a virucidal effect as the consequence of its interactions with the hemagglutinin (HA) 1 ectodomain of the envelope HA glycoprotein [18]. Importantly, synthetic PAC-A2 was observed to hinder IAV and IBV replication and determine a complete loss of infectivity of IV particles in virucidal assays, thus indicating PAC-A2 to be a bioactive phytochemical which plays an instrumental role in the overall anti-IV activity of this cranberry extract [18]. Consistently with this, we concluded that the interactions of PAC-A2 with HA1 and the subsequent alterations to the HA functions determined the loss of infectivity of IV particles. Here, we report for the first time that a novel formulation based on a mixture of four different plant extracts, each characterized by a very high content of dimeric A-type PACs (Appendix A), exerts a potent concentration-dependent inhibitory activity against different human respiratory viruses, including clinical isolates of SARS-CoV-2 and two endemic hCoVs. Likewise, as previously observed for IVs, the mechanism of SP4™’s anti-hCoVs activity involves the interaction of PAC-As with viral spike proteins and the subsequent impairment of spike-mediated adsorption to host cells.

Regarding hCoVs, a fraction derived from a cinnamon bark extract was reported to inhibit SARS-CoV replication. In this case, the fraction exerted a virucidal effect since inhibition was most effective when exposing the virus to the fraction before cell infection. Interestingly, of the different phytochemicals purified from the cinnamon fraction, PAC-A2 showed anti-SARS-CoV activity [20]. More recently, with respect to SARS-CoV-2, a library of more than 1000 plant-derived compounds was screened to identify agents able to prevent SARS-CoV-2 entry into target cells [21]. Procyanidin was among the four identified as entry inhibitors, and subsequent mechanistic studies indeed revealed its ability to exert a virucidal effect on SARS-CoV-2. Therefore, an interaction of procyanidin with the spike protein was hypothesized [21]. The results from the present study sustain that hypothesis, since the PAC-A-rich fraction of SP4™ was indeed shown to target the S protein of both hCoV-OC43 and SARS-CoV-2 and underlie its mode of action (Figure 7 and Figure 8). 

This general mechanism of the antiviral activity of PAC-As could result from the natural propensity of polyphenols to bind and aggregate proteins [22,23]. In this regard, it has been proposed that different types of chemical linkages, including hydrogen bonding, electrostatic interactions, and even covalent bonds, may contribute to the formation of complexes between protein and polyphenol [23]. Indeed, we observed that the alterations in the electrophoretic mobility of purified spike proteins following treatment with either SP4™ or F3 were resistant to boiling in the SDS sample buffer (Figure 7 and Figure 8). This supports the possibility that the exposure of purified viral glycoproteins to PAC-As may indeed result in the formation of covalent bonding between the two partners. The formation of both hydrogen and covalent bonds between the reactive quinone groups of dimeric PAC-As (Appendix A) [23] and the S protein may eventually lead to the extensive crosslinking of hCoVs glycoproteins. This would be expected to result in the smearing and eventual disappearance of the S protein bands, as observed in the electrophoretic mobility shift assay experiment with purified S (this study) and as previously seen with IV HA, HSV gB and gD glycoproteins [18,24]. Although the glycosylation of these envelope glycoproteins might affect the binding affinity of PAC-As to proteins [23], it should also be underlined that pomegranate polyphenols affect the infectivity of IV virions by interacting with HA and neuraminidase (NA) glycoproteins [25]. This observation suggests PACs to exert a certain degree of preference for binding to the abundant glycoproteins present on the envelope of respiratory viruses, such as those of hCoVs and IVs. That said, it remains to be established whether the observed alterations in the S proteins originate from the binding of SP4™’s PAC-As to specific domains of the viral protein or whether they simply “coat” the S protein’s surfaces. A future in silico docking simulation analysis will allow us to predict the docking propensity of the most active PAC-As components of SP4™, such as PAC-A1 and PAC-A2, to bind to the S protein, as well as their ability to bind to specific protein domains.

There are several examples of plant-derived natural products with BSA activity against human respiratory viruses that affect the initial interactions between viral envelope glycoproteins and host cell receptors [11,26]. For example, BanLec, a lectin isolated from banana fruits, and griffithsin, a mannose-specific lectin obtained from a red marine alga, have both been shown to inhibit the infectivity of both IV and hCoVs by interacting with envelope glycoproteins, thereby blocking attachment to host cell receptors [11]. Similarly, some plant-derived pentacyclic triterpenoids, such as the echinocystic and oleanolic acids, have been reported to block the attachment of IVs by interfering with HA [11]. Regarding the flavonoids, the flavonol isorhamnetin has been observed to inhibit the attachment and entry of SARS-CoV-2 [27]. However, it should be highlighted that, although SP4™ seems to act with a similar mechanism of action to the above natural-derived antivirals, it is characterized by some novelty elements worth highlighting. First, SP4™ is a mixture of four different plant extracts, intentionally chosen on the basis of their high natural content of PAC-As to especially increase the concentration of bioactive dimers (Appendix A). Second, in comparison to natural cranberry extracts characterized by high contents of PAC-As, SP4™ is more potent against respiratory viruses such as IV (the EC_50_ of SP4™ against IAV, as determined in this study, is 2.3 μg/mL (Table 1), while that of the potent cranberry extract Oximacro^®^ against the same IAV strain is 4.5 μg/mL [18]). Third, SP4™ is more soluble than pentacyclic triterpenoids and less cytotoxic than lectins, which could produce side effects in clinical applications [11]. Lasty, the production of SP4™ is characterized by its lower cost compared to extracts from a single plant source and endowed with similar PACs levels (see patent WO2023182971A1) [28].

A second point worth discussing pertains to the BSA activity of SP4™ (Figure 1) and its virucidal activity (Figure 6). In fact, although proven experimentally for hCoV-OC43 only (Figure 5 and Figure 6), it is very likely that the inhibitory activity of SP4™ against the different respiratory viruses tested in this study originates from the inhibition of virus attachment, as a consequence of alterations to the envelope glycoproteins essential for attachment and entry, thereby interfering with their binding partners on target cells. Thus, the BSA activity of SP4™ against common human respiratory viruses advocates for its potential application for the prevention of RTIs. In light of this hypothesis, it is possible to envisage SP4™, as a new attachment/entry inhibitor, facilitating the prevention of many RTIs through its capacity to inactivate the infectivity of a broad range of respiratory viruses. Formulations like SP4™ may also provide a source of antiviral agents able to block virus shedding and, thus, direct person-to-person transmission. From this perspective, the local application of SP4™ in the upper respiratory tract, administered in the form of either tablets/chewing gums, nose drops or through inhalation, would allow its active PAC-As to inactivate infecting viruses, thus preventing the start of RTIs. 

In this regard, the virucidal action of SP4™ could also permit its topical application in the upper respiratory tract instead of administration via the gastrointestinal route, thereby overcoming the limitations linked to the reduced bioavailability of PACs following their dietary consumption [29,30]. It is thought that the absorption of specific PACs through the intestinal barrier depends on their degree of polymerization (DP), which determines the poor absorption of polymeric PAC through the gut barrier and their elimination in the feces, only partially degraded by the gut microbiota [31,32]. Indeed, several studies in animals and humans have highlighted that only PAC dimers and, to a lesser extent, some trimers can be absorbed in the small intestine, while larger oligomers and polymers remain in the lumen and accumulate in the colon [32]. In fact, studies have confirmed the absorption of dimeric and trimeric PACs (obtained from different plant sources) through the intestinal epithelium and their presence in the plasma and urine [33,34], albeit with a 100-fold-reduced bioavailability compared to the epicatechin monomer [35]. Noteworthy, the gut absorption and urinary secretion of dimeric PAC-A2 was also measured in rodents orally administered with this A-type PAC [32]. Indeed, the adsorption of PAC-A2 (Appendix A) is of great importance, given that it represents a main active antiviral phytochemical of SP4™ (Appendix A and Table 3).

The excellent safety profile of many formulations of PAC-A-enriched plant extracts, such as the dried cranberry extracts widely used to reduce the risk of recurrent urinary tract infections (UTIs), provides further evidence supporting their use also in topical medications [14,36,37,38,39]. Indeed, the cytotoxicity values for the different synthetic PAC-As, tested in our study using a variety of human cell lines, were all very low (Table 3), supporting their potential safety for human administration. Interestingly, the antibacterial activity that SP4™ [16] could confer added further value to this phytocomplex through the prevention of respiratory tract bacterial superinfections, which may follow viral RTIs due to virus- and immune-mediated damage of the respiratory mucosa.

Lastly, our findings offer promise for the development of BSAs accessible to low-income countries. In fact, in many low- and middle-income countries, antiviral drugs are often beyond the financial reach of the people who need them most, or they are simply unavailable. Thus, the production of affordable, reliable, and high-quality low-cost antivirals is of great importance for the healthcare system of these countries. Being formulated from plant extracts recycled from food production (see patent WO2023182971A1) [28] renders SP4™ substantially cheaper (by approx. two-thirds) than cranberry extracts possessing the same PACs levels. This fact further supports its feasibility as an affordable BSA in low-income countries where the incidence of viral RTIs is still high. From this perspective, the development of phytocomplexes, such as SP4™, designed to include a high content of dimeric PAC-As, into new BSAs would not only be advantageous from an economic viewpoint, considering the more expensive purification procedures or chemical synthesis of specific PAC-As molecules, but it would also allow for the better exploitation of the synergistic and holistic effects of different bioactive PAC-As naturally present in plant-derived extracts [14,15,16]. In summary, PAC-A-rich phytocomplexes could offer suitable candidates for both the preclinical and clinical development of PAC-A-based BSAs, owing to the presence of the most active components contributing to their overall antiviral activity.

## 4. Materials and Methods

### 4.1. Compounds

SP4™ is a worldwide patented formulation (WO2023182971A1) composed of a mixture of extracts from four different plant sources: *Pinus pinaster* bark, *Vitis vinifera* grape seeds, *Arachis hypogaea* skins, and *Vacciniunm vitis-idaea* fruits and leaves. SP4™ contains a high content of polyphenols, flavonoids, anthocyanins, and, especially, PACs (precisely, 379.43 ± 12.44 mg/g) [16]. It was provided for this study by Calliero SpA (Moretta, Italy). The SP4™ powder and its derived fractions were resuspended in 100% dimethyl sulfoxide (DMSO). Synthetic PACs A1, A2, and A4 were purchased from Merck (Milan, Italy); synthetic PACs B1, B2, and B3 were obtained from Extrasynthese (Genay Cedex, France). All PACs were resuspended in 96% ethanol (*v*/*v*).

### 4.2. Purification and Chemical Characterization of SP4™ Bioactive Molecules

SP4™ was fractionated using the Biotage Seleckt Flash Chromatographic System (Biotage, Uppsala, Sweden). Briefly, SP4™ powder was dissolved in 70% (*w*/*v*) ethanol at a concentration of 100 mg/mL, and the extract was subjected to two sequential fractionation processes. For the first separation process (primary fractionation), we used a Biotage^®^ Sfär DLV column (inner diameter: 4 cm; length: 15 cm) packed with 50 g of Sephadex TM LH-20 resin (Merck; Darmstadt, Germany). Following a preconditioning phase with 70% (*v*/*v*) ethanol, 6 mL of 100 mg/mL SP4™ was loaded into the column. The first fraction (F1) was obtained by chromatographic separation carried out using 70% (*v*/*v*) ethanol; fraction 2 (F2) was obtained using a mixture of 2:2:1 (*v*/*v*/*v*) ethanol:methanol:H_2_O; and fraction 3 (F3) was obtained by fluxing 80% (*v*/*v*) acetone. The collected primary fractions F1–F3 were then dried using Rotavapor^®^ R-100 (Buchi, Cornaredo, Italy) for further fractionation and chemical analysis. F1 and F3 were further fractionated, leading to the formation of three secondary fractions from F1 (F1A, F1B, and F1C) and five secondary fractions from F3 (F3A, F3B, F3C, F3D, and F3E). This second fractionation was achieved by reverse chromatography to separate the different compounds based on their polarity. This process utilized a Biotage^®^ Sfär C18 D-Duo 100 Å 30 µm 50 g column (inner diameter: 4 cm; length: 15 cm) (Biotage, Sweden). After loading 100 mg/mL of either F1 or F3, chromatographic separation was achieved using a gradient flow composed of 0.1% (*v*/*v*) formic acid (solvent A) and acetonitrile acidified with 0.1% (*v*/*v*) formic acid (solvent B). The gradient started with a flow of solvent B at 5% (*v*/*v*) for 1 column volume (CV) in 2 min and gradually increased to 10% (*v*/*v*) in 6 min. for a total of 3 CV. Subsequently, the percentage of solvent B was raised to 50% (*v*/*v*) in 12 min (4 CV) and then to 95% (*v*/*v*) in 30 min (3 CV). This percentage was maintained for an additional 6 min (2 CV) to ensure the complete elution of the compounds from the chromatographic column.

The total PAC content (tPAC) was measured using the BL-DMAC assay based on the PAC-A2 standard [40]. Briefly, fractions were diluted in a PAC extraction buffer (75% (*v*/*v*) acetone with 0.5% (*v*/*v*) acetic acid) at a 1:10 (*w*/*v*) ratio. Each fraction was vortexed for 5 min and then placed in an ultrasound bath at room temperature for 30 min. The solution was further mixed on a swing plate for 1 h. Then, the samples were centrifuged at 5000× *g* for 5 min, and the supernatant was collected for tPAC quantification using the BL-DMAC assay [40]. All extractions and quantifications were performed in triplicate. For the quantification, 28 μL of each sample was incubated with 84 μL of 4-(dimethylamino)cinnamaldehyde (DMAC) reagent (0.1 mg/mL dissolved in 75% ethanol and acidified with 12% HCl). After the incubation period, absorbance was measured at 640 nm, as previously described [40]. The quantification was performed in triplicate within the linear range of the calibration curves (5–30 µg/mL) using pure PAC as the external standard. 

To identify and quantify the phytochemical compounds, we performed high-performance liquid chromatography–diode array detection–electro-spray ionization tandem mass spectrometry (HPLC-DAD-ESI-MS/MS), using a previously reported procedure [16,41]. We used an Agilent Technologies 1200 HPLC system coupled with a diode array detector (DAD) and a 6330 Series Ion Trap Mass Spectrometry (MS) System (Agilent Technologies, Santa Clara, CA, USA) featuring an electrospray ionization (ESI) source. Compounds were identified by comparing retention times (RT) and UV-VIS/MS spectra with authentic reference compounds or data from the literature. Quantification was achieved using calibration curves from pure standard injections. For anthocyanin analysis, we used a binary solvent system consisting of MilliQ H_2_O acidified with 10% (*v*/*v*) formic acid (solvent A) and 50% (*v*/*v*) methanol acidified with 10% (*v*/*v*) formic acid (solvent B). Elution involved a multistep linear gradient that started at 85% (*v*/*v*) solvent A and 15% (*v*/*v*) solvent B, shifting to 55% (*v*/*v*) solvent A and 45% (*v*/*v*) solvent B over 15 min, and then to 30% (*v*/*v*) solvent A and 70% (*v*/*v*) solvent B in 20 min. The initial solvent conditions were restored and maintained for an additional 10 min before injecting the next sample. The sample injection volume was 10 μL [41]. For other polyphenolic compounds (including PACs), the solvent system was MilliQ H_2_O acidified with 0.1% (*v*/*v*) formic acid (solvent A) and acetonitrile acidified with 0.1% (*v*/*v*) formic acid (solvent B). The gradient started at 90% (*v*/*v*) solvent A and 10% (*v*/*v*) solvent B for 5 min, increased to 45% (*v*/*v*) solvent A and 55% (*v*/*v*) solvent B over 25 min, and finally reached 30% (*v*/*v*) solvent A and 70% (*v*/*v*) solvent B over 25 min. The initial conditions were restored and held for 10 min before the next injection. The sample injection volume for these analyses was 5 μL. All analyses were performed in triplicate [16,41].

### 4.3. Cells and Viruses

The human adenocarcinoma alveolar basal epithelial A549 (ATCC CCL-185), the human laryngeal squamous carcinoma HEp-2 (ATCC CCL-23), the Madin-Darby canine kidney cell line MDCK (ATCC CCL-34), the human lung fibroblast cells MRC5 (ATCC CCL-171), the human lung adenocarcinoma cell line Calu-3 (ATCC HTB-55), the human ileocecal adenocarcinoma cell line HCT-8 (ATCC CCL-244), and the African Green monkey kidney cell line Vero E6 (ATCC CRL-1586) were all purchased from the American Type Culture Collection (ATCC, Manassas, VA, USA). HCT-8 cells were maintained in RPMI (Euroclone, Pero (MI), Italy), while the other cell lines were cultured in Dulbecco’s modified Eagle medium (DMEM, Euroclone). Growth media contained 10% (*v*/*v*) fetal bovine serum (FBS), 2 mM glutamine, 1 mM sodium pyruvate, 100 U/mL penicillin, and 100 μg/mL streptomycin sulfate (Euroclone).

Human coronavirus OC43 (hCoV-OC43, ATCC VR-1558) and human coronavirus 229E (hCoV-229E, ATCC VR-740) were purchased from ATCC and propagated and titrated as previously described in HCT-8 and MRC5 cells, respectively [42]. The SARS-CoV-2 Wuhan 1 isolate (SARS-CoV-2/human/ITA/CLIMVIB2/2020) was provided by the Virology Unit of Ospedale Luigi Sacco (Milan, Italy) (GenBank accession no. ON062195 MW000351.1). The SARS-CoV-2 Omicron BA.1 variant (GenBank accession no. ON062195) was provided by the Microbiology Unit of the University Hospital of Padova (Padova, Italy) [43]. Viral stocks of SARS-CoV-2 were produced and titrated by a plaque assay in Vero E6 cells. All experiments involving live SARS-CoV-2 were performed in compliance with the Italian Ministry of Health guidelines for biosafety level 3 (BSL-3) containment procedures in the approved laboratory of the Department of Molecular Medicine at the University of Padova.

The influenza A virus strain A/Puerto Rico/8/34 (IAV) and the Influenza B virus strain B/Lee/40 (IBV) were a generous gift from Arianna Loregian (University of Padova, Padova, Italy). IAV and IBV propagation and titration by plaque assay were performed on MDCK cells in the presence of 2 μg/mL of TPCK-treated trypsin from bovine pancreas (Merck) and 0.14% (*v*/*v*) of bovine serum albumin (Merck) [18]. Respiratory syncytial virus strain Long A (RSV-A, ATCC VR-26) was purchased from ATCC and produced and titrated in HEp-2 cells as previously described [44].

### 4.4. Antiviral Assays

The antiviral activities of SP4™, its fractions, or the purified PACs were evaluated against hCoV-OC43 and hCOV-229E by means of the focus-forming reduction assay (FFRA) procedure. For hCoV-OC43, HCT-8 cells were seeded on 96-well plates (30,000 cells/well) and, after 24 h, treated with different concentrations of the compounds 1 h prior to and during infection with hCoV-OC43 (50 PFU/well). After virus adsorption (2 h at 33 °C), the infected cell cultures were incubated in medium containing the corresponding compounds plus 1% (*w*/*v*) methylcellulose (Merck) and 1% (*v*/*v*) FBS. At 72 h post infection (p.i.), cell monolayers were fixed and subjected to indirect immunoperoxidase staining (IPA) with an mAb against the hCoV-OC43 nucleoprotein (N) (Millipore, clone 542-D7; Burlington, MA, USA) (diluted 1:100). Viral foci were microscopically counted, and the mean plaque counts for each drug concentration was converted in viral titer (PFU/mL). GraphPad Prism software version 8.0 was used to determine the concentration of compounds producing 50 and 90% reductions in viral titers (EC_50_ and EC_90_). For hCoV-229E, MRC5 cells were seeded on 96-well plates (40,000 cells/well), and after 24 h they were exposed to increasing concentrations of SP4™ and then infected 1 h later with hCoV-229E (50 PFU/well). After 2 h at 33 °C, viral inocula were removed, and the cells were incubated in DMEM containing SP4™, 1.5% (*w*/*v*) methylcellulose and 1% (*v*/*v*) FBS. At 48 h p.i., infected MRC5 cell monolayers were fixed and stained as described above using a polyclonal antibody (pAb) against the hCoV-229E N protein (Sino Biological, 40640-T62; Beijing, China). 

The antiviral activity of SP4™ against IAV and IBV was determined using the virus yield reduction assay (VRA) in A549 cells. Briefly, A549 cells seeded on 48-well plates (120,000 cells/well) were treated with increasing concentrations of SP4™ and then infected with IAV or IBV at an multiplicity of infection (MOI) of 0.001 PFU/cell. After virus adsorption (1 h at 37 °C), the cells were incubated in medium containing SP4™, 2 μg/mL of TPCK-treated trypsin and 0.14% (*v*/*v*) of bovine serum albumin. At 48 h p.i., the cell supernatants were harvested, and the IV yields were titrated on MDCK cells as previously described [18].

The antiviral activity against RSV-A was determined by VRA on A549 cells treated with increasing concentrations of SP4™ before, during, and after infection with RSV-A at an MOI of 0.01 for 2 h at 37 °C. After virus adsorption, the viral inoculum was removed, and the infected cell monolayers were incubated in medium supplemented with 2% (*v*/*v*) FBS and SP4™. At 72 h p.i., the supernatants from RSV-infected cells were collected and titrated on HEp-2 cells [34].

The anti-SARS-CoV-2 activity of SP4™ was measured by VRA on Calu-3 cells as previously described [42]. Briefly, Calu-3 cells seeded on 96-well plates (27,500 cells/well) were exposed to increasing concentrations of SP4™ 1 h prior to infection with SARS-CoV-2 Wuhan or Omicron BA.1 at an MOI of 0.01. Following 1 h of adsorption at 37 °C, the virus inoculum was removed and replaced with fresh medium supplemented with the tested compound or vehicle. At 48 h p.i., cell supernatants were harvested, and the SARS-CoV-2 yields were titrated by a plaque assay on Vero E6 cells. 

For the time-of-drug-addition experiments, HCT-8 cells were seeded on a 96-well plate as for FFRA. The following day, as summarized in the scheme in Figure 4A, the cell monolayers were treated: for 1 h with different concentrations of SP4™ before infection with hCoV-OC43 (from −1 to 0 h, i.e., pre-treatment; Pre-T), during infection (from 0 to 2 h, i.e., co-treatment; Co-T), or after viral infection (from 2 to 72 h p.i., i.e., post-treatment; Post-T). Then, at 72 h p.i., the viral foci were immunostained as described above. 

To differentiate viral attachment and entry, temperature-shift assays were performed as previously described [18,45]. Briefly, HCT-8 cells were seeded on a 96-well plate as for FFRA, and after 24 h, the cultures were chilled on ice for 20 min and then washed three times with cold medium. For the attachment assay, chilled HCT-8 cell monolayers were infected with hCoV-OC43 (50 PFU/well) at 4 °C for 2 h in the presence of different concentrations of SP4™ or its fraction 3. After virus adsorption, infected cell monolayers were washed three times with warm medium and incubated with medium supplemented with 1% (*w*/*v*) methylcellulose and 1% (*v*/*v*) FBS for 72 h at 33 °C. The viral foci were then immunostained as described above. For the entry assay, HCT-8 cell monolayers were infected with hCoV-OC43 (50 PFU/well) for 2 h at 4 °C. Thereafter, the infected cells were washed three times with cold medium, treated with different concentrations of SP4™ or fraction 3, and incubated for 2 h at 33 °C to allow hCoV-OC43 entry. Subsequently, the cell monolayers were exposed to cold acidic glycine buffer (100 mM glycine, 150 mM NaCl, pH 3) for 30 s to remove the adsorbed virions and washed three times with warm medium. The infected cell monolayers were then incubated with medium supplemented with 1% (*w*/*v*) methylcellulose and 1% (*v*/*v*) FBS for 72 h at 33 °C, and the viral foci were measured by FFRA as described above.

To measure the effects of SP4™ and fraction 3 on hCoV-OC43 infectivity, a virucidal assay was performed by incubating 10^5^ PFU/mL with SP4™ or fraction 3 (15 μg/mL) in serum-free medium for 0, 60, 120, or 180 min at either 4 °C or at 33 °C. Then, viral suspensions were diluted to reduce the compound concentration below that required for antiviral activity, and residual hCoV-OC43 infectivity was measured by FFRAs as described above.

### 4.5. Cytotoxicity Assay

HCT-8, MRC5, A549, and Calu-3 cells seeded on 96-well plates 24 h before the assay were exposed to increasing concentrations of compounds or the vehicle as a control. Cytotoxicity was evaluated after 48 h for MRC5 and Calu-3 cells or after 72 h for A549 and HCT-8 cells, using the CellTiter-Glo assay (Promega, Milan, Italy) or the ATPlite assay (PerkinElmer, Milan, Italy). The cytotoxicity of the tested compounds was expressed as cytotoxic concentration (CC_50_).

### 4.6. Analysis of Proteins

For the electrophoretic mobility shift assay experiments with purified S glycoproteins [18,24], 3 μg of purified recombinant hCoV-OC43 spike glycoprotein (S1 + S2 ECD, His Tag) (Sino Biological, 40607-V08B) produced in baculovirus-infected insect cells or 1 μg of SARS-CoV-2 B.1.1.529 (Omicron BA.1) (S1 + S2 ECD, His Tag) (Sino Biological, 40589-V08H26) expressed in HEK293 cells were incubated at 33° or 37 °C for different times with the compounds. Then, the mixtures were fractionated through 4–15% (*w*/*v*) SDS-PAGE (BioRad; Hercules, CA, USA) that had been stained with Coomassie blue. Electrophoretic protein patterns were analyzed by ChemiDoc (BioRad). Recombinant S protein preparations were also analyzed by immunoblotting as previously described [18], using either the anti-hCoV-OC43 S1 rabbit polyclonal antibody (pAb) (CABT-CS063; Creative Diagnostic; Shirley, NY, USA) (diluted 1:500) or the anti-SARS-CoV-2 spike protein (S1) rabbit monoclonal antibody (mAb) (99423; Cell Signaling Technologies; Danvers, MA, USA) (diluted 1:1000). Immunocomplexes were then detected with goat anti-rabbit Ig Ab or goat anti-mouse Ig Ab conjugated to horseradish peroxidase (Life Technologies; Carlsbad, CA, USA) and visualized by enhanced chemiluminescence (Clarity Western ECL Substrate, BioRad).

### 4.7. Statistical Analysis

All data were generated from at least three independent experiments performed in triplicate. Statistical tests were performed using GraphPad Prism version 8.0 (GraphPad Software; Boston, MA, USA). The results from the antiviral assays are presented as means ± SD. Data from the time-of-drug-addition experiments and the virucidal assays were analyzed by a two-way ANOVA, followed by Dunnett’s multiple comparison test to create confidence intervals for the differences between the mean of each factor (timing or concentration) of the treated samples and the mean of a control group (CV) and considered to be statistically significant for *p* values ≤ 0.05. Significance was set at * (*p* < 0.05), ** (*p* < 0.01), *** (*p* < 0.001), or **** (*p* < 0.0001).

## 5. Conclusions

In conclusion, the results of this study indicate that the ability of the new SP4™ phytocomplex to inhibit the replication of major human respiratory viruses belonging to different virus families could be exploited to develop strategies for the prevention of RTIs. The in vitro BSA activity of SP4™ advocates for further evaluation in safety and efficacy studies in animal models of human viral RTIs, to determine whether it should be proposed for the development of effective BSAs which can be rapidly deployed against current and future emerging respiratory viruses.

## Figures and Tables

**Figure 1 ijms-25-07370-f001:**
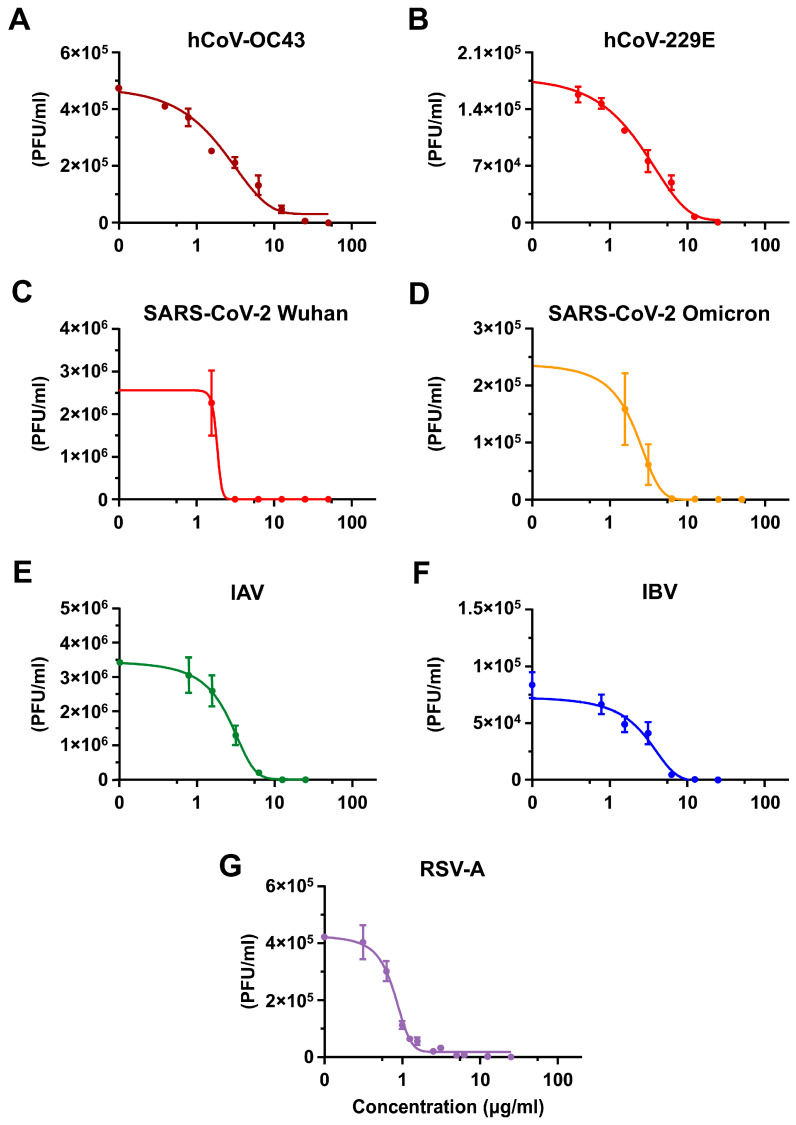
SP4™ inhibits the replication of hCoV-OC43, hCoV-229E, SARS-CoV-2, IAV, IBV, and RSV-A. (**A**,**B**) Focus forming reduction assays (FFRAs) were performed in HCT-8 and MRC5 cells infected with coronaviruses hCoV-OC43 (**A**) or hCoV-229E (**B**), respectively, and treated with different concentrations of SP4™ 1 h before, during, and post infection. At 72 h p.i., the viral foci were microscopically counted and converted into viral titer (plaque forming unit (PFU)/mL). (**C**–**G**) Virus yield reduction assays (VRAs) were performed in Calu-3 cells infected with SARS-CoV-2 Wuhan (**C**) or the Omicron BA.1 variant (**D**) and in A549 cells infected with influenza A (IAV) (**E**), influenza B (IBV) (**F**), or respiratory syncytial (RSV-A) viruses (**G**). Infected cell monolayers were treated throughout the experiment with different concentrations of SP4™. At 48 h p.i. for SARS-CoV-2 and IVs and at 72 h p.i. for RSV-A, cell supernatants were harvested and titrated by the plaque assay, as described in the Material and Methods. The data shown are the means ± standard deviations (SDs) (error bars) of *n* = 3 independent experiments performed in triplicate.

**Figure 2 ijms-25-07370-f002:**
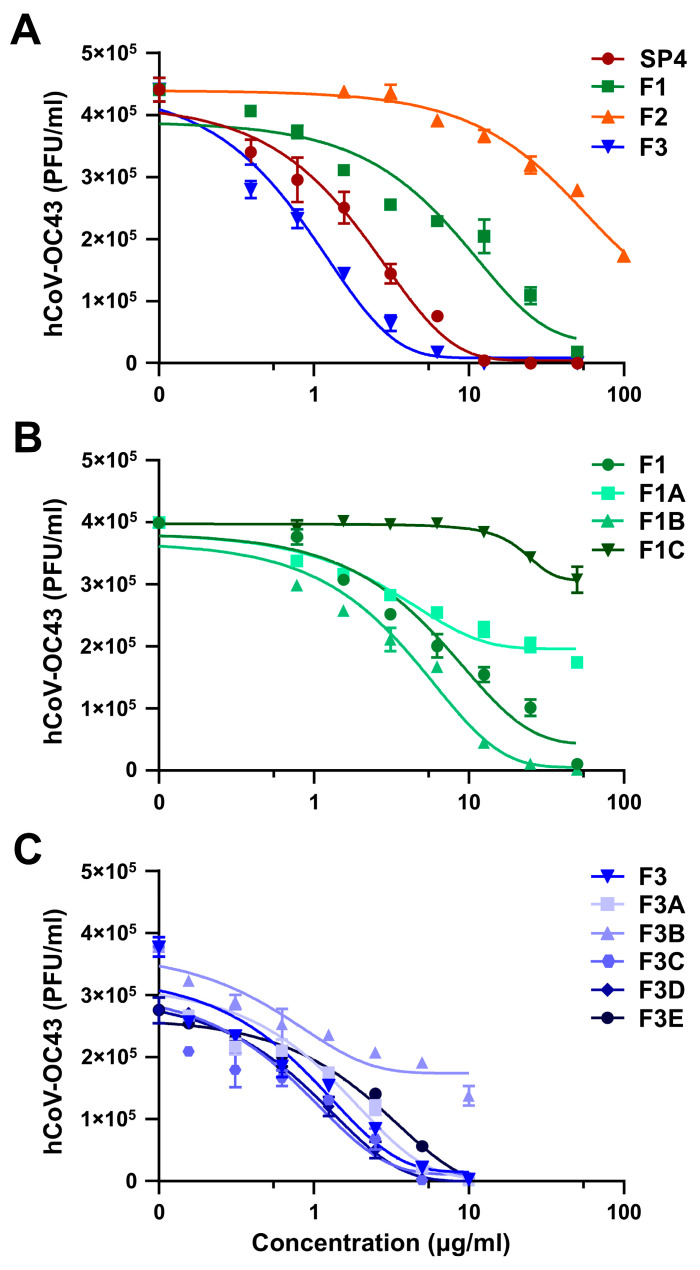
Anti-hCoV activity of the different fractions of SP4™. (**A**) FFRAs were performed in HCT-8 cells infected with hCoV-OC43 and exposed to different concentration of the primary fractions F1-F3. At 72 h p.i., the viral foci were microscopically counted and converted into viral titer (PFU/mL). (**B**,**C**) The secondary fractions obtained from the fractionation of F1 (F1A-C, (**B**)) or F3 (F3A-E, (**C**)) were analyzed by FFRAs in hCoV-OC43-infected HCT-8 cells treated with different concentrations of the samples. The results are shown as the means ± SD of *n* = 3 independent experiments performed in triplicate.

**Figure 3 ijms-25-07370-f003:**
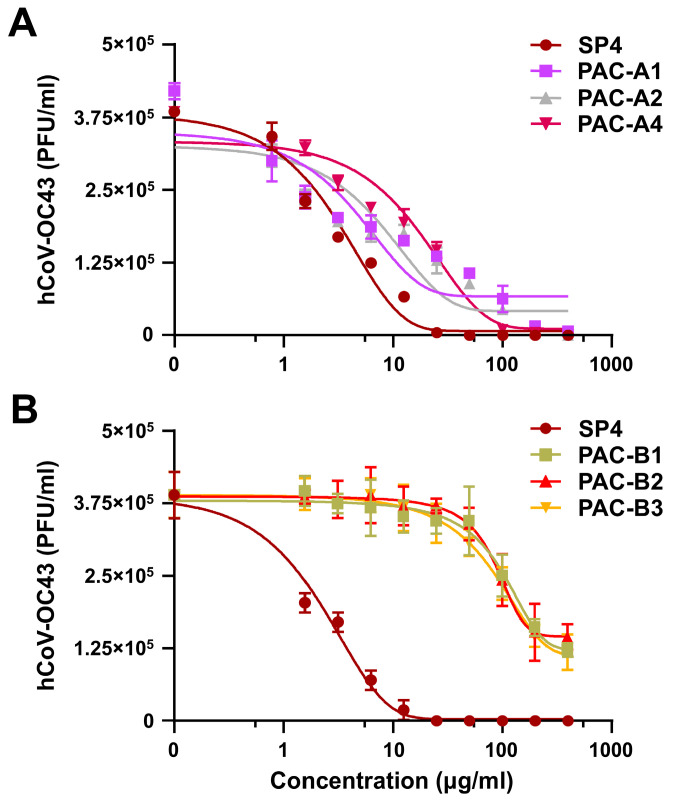
Synthetic dimeric PAC-As inhibit hCoV replication. (**A**,**B**) FFRAs were carried out in HCT-8 cells infected with hCoV-OC43 and treated with different concentrations of synthetic PAC-As (**A**) or PAC-Bs (**B**), present throughout the experiment. At 72 h p.i., the viral foci were microscopically counted and converted into viral titer (PFU/mL). The reported data are the means ± SDs of *n* = 3 independent experiments performed in triplicate.

**Figure 4 ijms-25-07370-f004:**
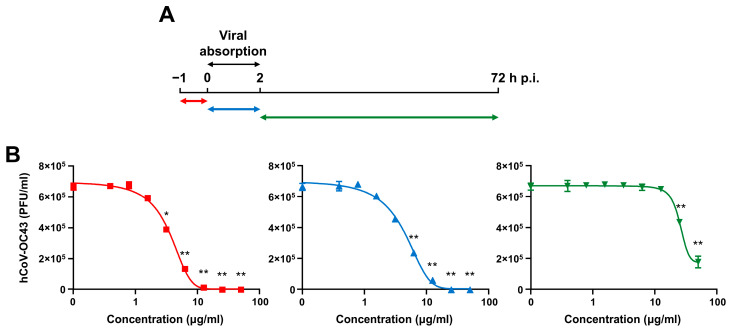
SP4™ acts during an early stage of the hCoV replicative cycle. (**A**) Schematic summary of the time-of-drug-addition experiment. HCT-8 cell monolayers were exposed to different concentrations of SP4™ 1 h prior to hCoV-OC43 infection (from −1 to 0 h; pre-treatment; red line), or during hCoV-OC43 infection (from 0 to 2 h p.i.; co-treatment; blue line), or after hCoV-OC43-infection (from 2 to 72 h p.i.; post-treatment; green line). (**B**) At 72 h p.i., hCoV-OC43 replication was assessed by FFRA, and the viral foci were microscopically counted and plotted as PFU/mL. The data shown are the means ± SD (error bars) of three independent experiments performed in triplicate and analyzed by a two-way ANOVA followed by Dunnett’s multiple comparison test. * *p* < 0.05; ** *p* < 0.01.

**Figure 5 ijms-25-07370-f005:**
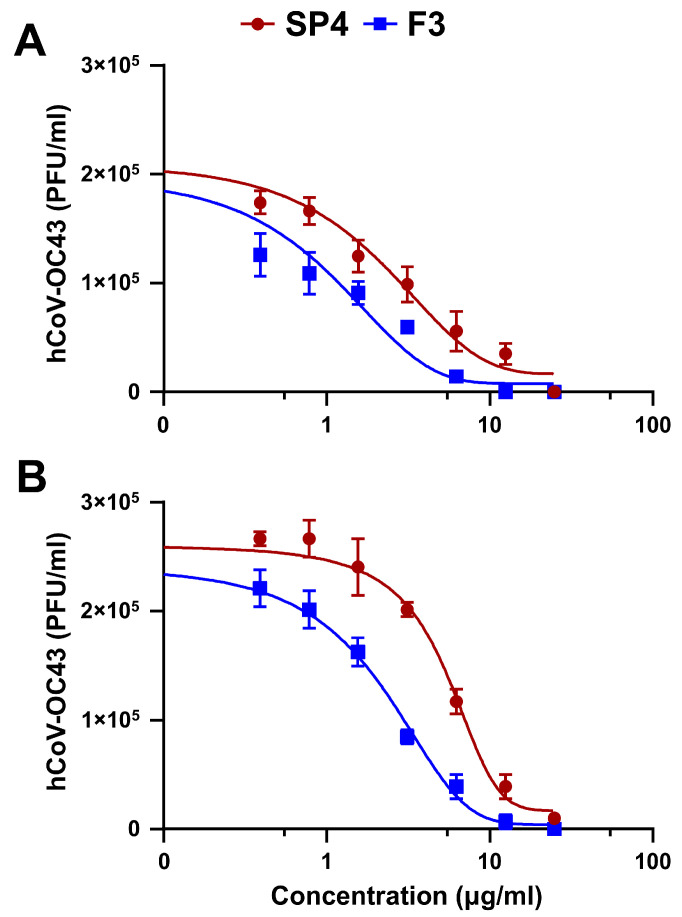
SP4™ prevents hCoV-OC43 attachment and entry. (**A**) For the attachment assay, prechilled HCT-8 cell monolayers were infected with hCoV-OC43 (50 PFU/well) at 4 °C for 2 h in the presence of different concentrations of SP4™ or F3, ranging from 0.39 to 25 μg/mL. Following viral adsorption, the infected cell monolayers were washed three times with medium, overlaid with growth medium supplemented with 1% methylcellulose, and incubated at 33 °C. At 72 h p.i., viral foci were immunostained and microscopically counted. The results shown are the means ± SD from *n* = 3 independent experiments performed in triplicate. (**B**) For the entry assay, prechilled HCT-8 cells were infected with hCoV-OC43 (50 PFU/well) for 2 h at 4 °C to allow virion attachment to the cells. Thereafter, the cells were washed with medium and treated with different amounts of SP4™ or F3 for 2 h at 33 °C, prior to the inactivation of bound extracellular virus with acidic glycine buffer for 30 s at room temperature. After further washings, cell monolayers were incubated for 72 h at 33 °C, and the viral foci were detected as in the attachment assay. The results shown are the means ± SD of *n* = 3 independent experiments performed in triplicate.

**Figure 6 ijms-25-07370-f006:**
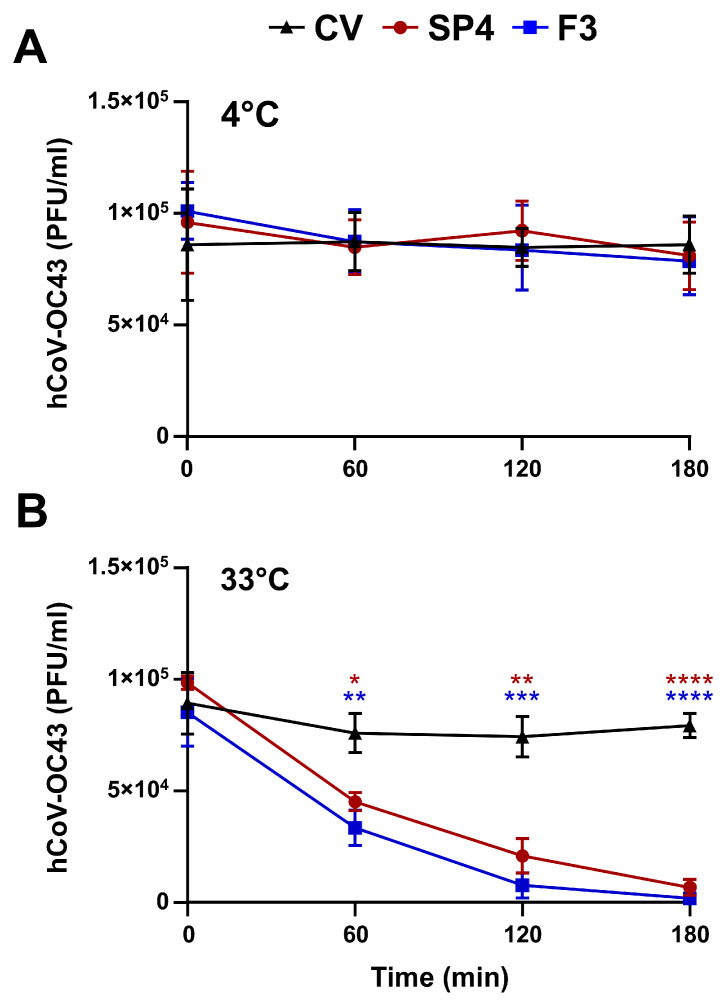
SP4™ abrogates the infectivity of hCoV-OC43 virions. hCoV-OC43 (10^5^ PFU/mL) was incubated at either 4 °C (**A**) or at 33 °C (**B**) for different times in the absence (control virus (CV)) or presence of 15 μg/mL SP4™ or F3. After incubation, the samples were diluted to reduce the concentration of SP4™ or fraction 3 below the concentration at which hCoV-OC43 attachment is inhibited (0.1 μg/mL). The foci were microscopically counted at 72 h p.i., and the mean number of foci counts was expressed as PFU/mL. The results are representative of *n* = 3 independent experiments performed in triplicate. The data were analyzed by a two-way ANOVA, followed by Dunnett’s multiple comparison test. A statistical analysis was performed by comparing the treated samples with the CV control for each condition. * *p* < 0.05; ** *p* < 0.01; *** *p* < 0.001; and **** *p* < 0.0001.

**Figure 7 ijms-25-07370-f007:**
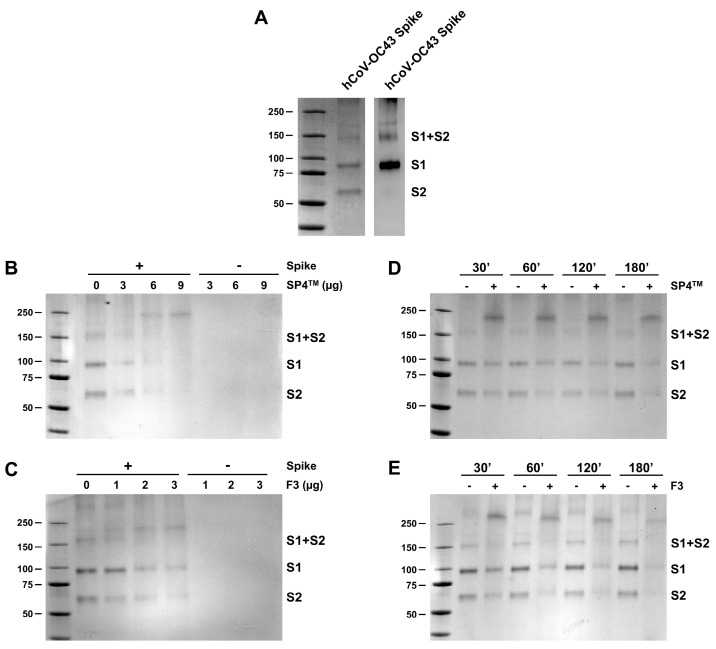
SP4™ interacts with the purified spike (S) protein of hCoV-OC43. (**A**) SDS-PAGE and the immunoblot analysis of the recombinant S protein of hCoV-OC43. The purified recombinant S protein preparation was analyzed by 4–15% SDS-PAGE, and the gels were either stained with Coomassie blue (left panel) or analyzed by immunoblotting with an anti-S (S1 subunit) polyclonal antibody (pAb) (right panel). Left panel: Coomassie blue-stained gel of 3 μg of the S protein preparation fractionated in the 160 kilodaltons (kDa) S1 + S2 component, the 95 kDa S1 subunit, and the 60 kDa S2 subunit. Right panel: Immunoblot analysis of 500 ng of the S protein preparation in which the S1 + S2 and the S1 components had been stained with anti-S1 monoclonal antibody (mAb). (**B**–**E**) SP4™ and F3 interact with the recombinant S protein of hCoV-OC43. (**B**,**C**) The purified recombinant S protein preparation (3 μg) was incubated at 33 °C with DMSO as a control or increasing amounts of either SP4™ (**B**) or F3 (**C**) for 3 h. The mixtures were then analyzed by 4–15% SDS–PAGE. (**D**,**E**) The recombinant S protein preparation (3 μg) was incubated at 33 °C for 30, 60, 120, and 180 min with DMSO or 9 μg of SP4™ (**D**) or 3 μg of F3 (**E**). At the indicated timepoints, the mixtures were analyzed by 4–15% SDS–PAGE. The gels were then stained with Coomassie blue, and images were acquired by ChemiDoc. The molecular weight markers are indicated in kilodaltons.

**Figure 8 ijms-25-07370-f008:**
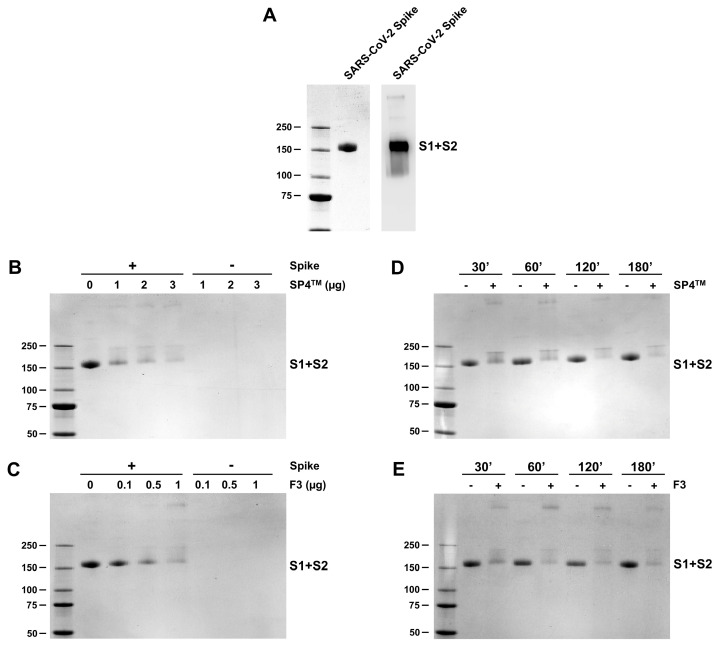
SP4™ targets the purified S protein of SARS-CoV-2. (**A**) SDS-PAGE and immunoblot analysis of recombinant SARS-CoV-2 (Omicron BA.1) S protein preparation. The purified S protein preparation (S1 + S2) was fractionated by 4–15% SDS-PAGE and either stained with Coomassie blue (left panel) or analyzed by immunoblotting with an anti-S1 mAb (right panel). Left panel: Coomassie blue-stained gel of 3 μg of the recombinant S preparation resolved as the 160 kDa S1 + S2 protein band. Right panel: Immunoblot analysis of 500 ng of the same sample as in the left panel. (**B**–**E**) SP4™ and F3 interact with the S protein of SARS-CoV-2. (**B**,**C**) The purified recombinant S protein preparation (1 μg) was incubated with DMSO as a control or increasing concentrations of SP4™ (**B**) or F3 (**C**) for 3 h at 37 °C; the mixtures were then analyzed by 4–15% SDS–PAGE. (**D**,**E**) Recombinant S protein aliquots (1 μg) were incubated at 37 °C for 30, 60, 120, and 180 min with DMSO or 3 μg of SP4™ (**D**) or 1 μg of fraction 3 (**E**). At the indicated incubation timepoint, the mixtures were analyzed by 4–15% SDS-PAGE. the Gels were then stained with Coomassie blue, and images were acquired by ChemiDoc. The molecular weight markers are indicated in kilodaltons.

**Table 1 ijms-25-07370-t001:** Broad-spectrum antiviral activity of SP4™ against human respiratory viruses.

Virus	Cell Line	EC_50_ (μg/mL) ^1^	EC_90_ (μg/mL) ^2^	CC_50_ (μg/mL) ^3^	SI ^4^
hCoV-OC43	HCT-8	2.20 ± 0.59	9.51 ± 1.76	139.08 ± 5.61	63.2
hCoV-229E	MRC5	2.52 ± 0.02	9.12 ± 0.04	296.95 ± 4.92	117.8
Wuhan SARS-CoV-2	Calu-3	2.10 ± 0.12	2.89 ± 0.03	>50	>23.8
Omicron SARS-CoV-2	Calu-3	2.10 ± 0.10	4.17 ± 0.76	>50	>23.8
IAV	A549	2.61 ± 0.08	5.26 ± 0.26	115.51 ± 7.36	44.3
IBV	A549	2.09 ± 0.69	5.64 ± 0.12	115.51 ± 7.36	55.3
RSV-A	A549	0.85 ± 0.04	1.45 ± 0.02	115.51 ± 7.36	135.9

^1^ 50% effective concentration (EC_50_), the compound concentration inhibiting 50% of virus replication, as determined against hCoV-OC43 by FFRAs in HCT-8 cells, hCoV-229E by FFRAs in MRC5 cells, SARS-CoV-2 by VRAs in Calu-3 cells, IAV and IBV by VRAs in A549 cells, and RSV-A by VRAs in A549 cells. ^2^ 90% effective concentration (EC_90_), the compound concentration inhibiting 90% of virus replication, as determined against hCoV-OC43 by FFRA in HCT-8 cells, hCoV-229E by FFRA in MRC-5 cells, SARS-CoV-2 by VRA in Calu-3 cells, IAV and IBV by VRA in A549 cells, and RSV-A by VRA in A549 cells. ^3^ 50% cytotoxic concentration (CC_50_), the compound concentration producing 50% of cytotoxicity, as determined by cell viability assays in HCT-8, MRC-5, Calu-3, or A549 cells. The reported values are the means ± SD of data derived from three experiments performed in triplicate. ^4^ SI, selectivity index determined as the ratio of CC_50_ to EC_50_.

**Table 2 ijms-25-07370-t002:** Antiviral activity of the fractions derived from SP4™.

Fraction	EC_50_ (μg/mL) ^1^	EC_90_ (μg/mL) ^2^	CC_50_ (μg/mL) ^3^	SI ^4^
1	5.01 ± 0.24	31.44 ± 2.43	>400	>79.84
1A	19.09 ± 3.62	>50	>400	20.95
1B	4.11 ± 0.24	14.10 ± 0.97	230.35 ± 8.74	56.05
1C	>50	>50	>400	8
2	60.39 ± 7.01	>100	>400	>6.62
3	0.78 ± 0.12	2.43 ± 0.72	49.84 ± 2.5	63.9
3A	1.03 ± 0.12	4.61 ± 0.18	43.63 ± 1.19	42.4
3B	3.23 ± 0.16	>10	>400	123.8
3C	0.31 ± 0.15	3.26 ± 0.25	53.72 ± 2.68	173.3
3D	1.02 ± 0.05	3.11 ± 0.18	48.62 ± 3.83	47.7
3E	2.21 ± 0.05	7.13 ± 0.25	129.36 ± 1.99	58.5

^1^ EC_50_, the compound concentration inhibiting 50% of virus replication, as determined against hCoV-OC43 by FFRAs in HCT-8 cells. The reported values are the means ± SD of data derived from three experiments performed in triplicate. ^2^ EC_90_, the compound concentration inhibiting 90% of virus replication, as determined against hCoV-OC43 by FFRAs in HCT-8 cells. The reported values are the means ± SD of data derived from three experiments performed in triplicate. ^3^ CC_50_, the compound concentration producing 50% of cytotoxicity, as determined by cell viability assays performed in HCT-8 cells. The reported values are the means ± SD of data derived from three experiments performed in triplicate. ^4^ SI, selectivity index determined as the ratio of CC_50_ to EC_50_.

**Table 3 ijms-25-07370-t003:** Antiviral activity of synthetic dimeric PACs against hCoV-OC43.

PAC	EC_50_ (μg/mL) ^1^	EC_90_ (μg/mL) ^2^	CC_50_ (μg/mL) ^3^	SI ^4^
A1	4.10 ± 0.71	139.35 ± 2.06	>400	>97.56
A2	6.58 ± 0.40	100.74 ± 0.66	>400	>60.79
A4	14.57 ± 1.39	68.39 ±0.92	>400	>27.45
B1	167.02 ± 6.51	>400	>400	>2.39
B2	142.41 ± 8.93	>400	>400	>2.81
B3	134.92 ± 4.14	>400	>400	>2.96

^1^ EC_50_, the compound concentration inhibiting 50% of virus replication, as determined against hCoV-OC43 by FFRAs in HCT-8 cells. The reported values are the means ± SDs of data derived from three experiments performed in triplicate. ^2^ EC_90_, the compound concentration inhibiting 90% of virus replication, as determined against hCoV-OC43 by FFRAs in HCT-8 cells. The reported values are the means ± SDs of data derived from three experiments performed in triplicate. ^3^ CC_50_, the compound concentration producing 50% of cytotoxicity, as determined by cell viability assays in HCT-8 cells. The reported values are the means ± SD of data derived from three experiments in triplicate. ^4^ SI, selectivity index determined as the ratio of CC_50_ to EC_50_.

## Data Availability

The data presented in this study are available upon request from the corresponding author.

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
