# Peer review of "The Novel A-Type Proanthocyanidin-Rich Phytocomplex SP4™ Acts as a Broad-Spectrum Antiviral Agent against Human Respiratory Viruses"

_ijms, 2024, doi:10.3390/ijms25137370_

Round 1

Reviewer 1 Report

Comments and Suggestions for Authors
  1. Increase the number of experimental replicates to strengthen the statistical validity of the findings.

  2. Provide more detailed data on the interaction between SP4™ and viral spike proteins, possibly through in silico docking studies or additional biophysical analyses.

  3. Expand the comparison with existing literature on natural product-derived broad-spectrum antivirals to better highlight the novelty of SP4™.

  4. Include additional information on the safety and toxicity profile of SP4™ to assess its potential for therapeutic use.

  5. Simplify complex figures and provide higher resolution images for better data interpretation.
  6. Include a detailed description of the statistical methods, tests used, and significance levels.

  7. Provide more comprehensive details on the purification and characterization of SP4™ to facilitate replication of the study.

Comments on the Quality of English Language

The manuscript exhibits a generally proficient use of English, with clear scientific communication and appropriate technical vocabulary. However, there are occasional grammatical errors and awkward phrasings that detract from the overall clarity. The language could benefit from further refinement by a professional editor or native English speaker to ensure consistency and enhance readability.

Reviewer 2 Report

Comments and Suggestions for Authors

I think the manuscript should be condensed. For example, the values ​​of EC50, CC50, etc. should not be duplicated in the text (see paragraph before Table 1) as they are already shown in the tables and should be discussed briefly in a general form in the text.

I recommend moving Table 2 to the Appendix and instead inserting a table with only the main components.

The limitations of using SP4 should be discussed.

Indeed, most attention in the paper was on respiratory tract infections. However, given the low bioavailability of PACs, the GI tract constitutes the major site of the biological actions of intact PACs (Cires et  al. 2016).

Additionally, the microbiota metabolizes PACs generating multiple compounds that can be absorbed and exert systemic actions.

It is widely accepted that only monomers and dimeric PACs can be absorbed at the GI tract. However, the bioavailability of PAC dimers is 100-times lower compared to that of monomeric flavan-3-ols (Holt et  al. 2002; Wiese et  al. 2015).

Oligomeric and polymeric PACs are not bioavailable and are found as intact parent compounds throughout the GI tract and in feces (Choy et  al. 2013; Ou and Gu 2014; Rios et  al. 2002; Wiese et  al. 2015).

Thus, the discussion should be modified to take these issues into account.

It should also be noted that some proanthocyanidins can alter biological activity even if they are not cytotoxic. For example, A1 is a PKC inhibitor.

Round 2

Reviewer 1 Report

Comments and Suggestions for Authors
  1. Druggability of Natural Products: The article highlights the potential of natural products as a source for broad-spectrum antiviral agents. However, the druggability of these natural compounds needs to be carefully evaluated. Early assessments of druggability should be conducted to ensure the compounds have the necessary properties for pharmaceutical development.

  2. Chemical Structures: The authors should provide the chemical structures of the representative bioactive compounds within the manuscript. This information is crucial for understanding the molecular basis of their antiviral activity and for further synthetic optimization.

  3. Pharmacokinetic Studies: It is recommended that the authors conduct pharmacokinetic studies of the preferred molecules in animal models. Understanding the absorption, distribution, metabolism, and excretion of these compounds is essential for assessing their potential as viable drug candidates.

  4. Mechanism of Action: Although the hypothesis of SP4TM interacting with viral spike proteins is proposed, more research is needed to elucidate the precise mechanism of action. Detailed mechanistic studies will strengthen the findings and provide insights into how these compounds could be further developed as therapeutic agents.

  5. Safety and Toxicology: The manuscript mentions the safety of SP4TM but lacks comprehensive toxicological data or an assessment of long-term safety. A thorough toxicological evaluation is necessary to ensure the compound's safety for human use.

In summary, while the study presents promising preliminary findings, further work is needed to address the druggability, detailed mechanism of action, and comprehensive safety profile of the natural product compounds presented.

Reviewer 2 Report

Comments and Suggestions for Authors

The paper could be accepted in present form.

Author Response

We thank Reviewer 2 for the positive comment on the revision of our manuscript. Her/his suggestions and comments certainly have improved the quality of the manuscript. Thank you again for your valuable and constructive review.